# Investigation of the Influence of Gas Turbine Power Stations on the Quality of Electric Energy in the Associated Petroleum Gas Utilization

Anton Petrochenkov [1,2,*], Aleksandr Romodin [1], Dmitriy Leyzgold [1], Andrei Kokorev [1], Aleksandr Kokorev [1], Aleksandr Lyakhomskii [2], Evgenia Perfil'eva [2], Yuri Gagarin [3], Ruslan Shapranov [3], Pavel Brusnitcin [3] and Pavel Ilyushin [4]

[1] Electrical Engineering Faculty, Perm National Research Polytechnic University, 614990 Perm, Russia; romodin_av@pstu.ru (A.R.); leizgold_du@pstu.ru (D.L.); kaa19.77@mail.ru (A.K.); kok881@yandex.ru (A.K.)
[2] College of Mining, National University of Science and Technology MISIS, 119049 Moscow, Russia; lav5723807@mail.ru (A.L.); mggu.eegp@mail.ru (E.P.)
[3] Sputnic-Komplektatsia LLC, 614990 Perm, Russia; gua@sputnic.ru (Y.G.); r.shapranov@sputnic.ru (R.S.); p.brusnitsyn@sputnic.ru (P.B.)
[4] Research and Educational Center for Geology, Oil and Gaz Fields Development, 614990 Perm, Russia; ilushin-pavel@yandex.ru
[*] Correspondence: pab@msa.pstu.ru; Tel.: +7-3422391821

**Abstract:** The problem of the quality of electric energy in the utilization of petroleum gas is considered. The article presents the results of the development of a mathematical description of power supply systems with gas turbine power stations based on two-shaft gas turbine units. The typical power distribution scheme of a gas turbine power station is given. The joint analysis of the generation modes of the gas turbine power station and the detected deviations of the power quality indicators values was carried out. The influence of the used mode on the power quality indicators is determined. As a result, the factors of operation of a gas turbine power station that affects power quality are identified, and recommendations for their elimination are given.

**Keywords:** gas turbine power station; power supply system simulation; load module; imbalance of power; power quality indicators; local mode; parallel networking mode

## 1. Introduction

The United Nations currently proposes the blueprint of the Sustainable Development Goals (SDG). According to the authors, the relevance of SDG achievement is due to any modern oil and gas production enterprise. In particular, "Goal 12: Responsible Consumption and Production" [1] can be achieved by increasing the efficiency of the associated petroleum gas (APG) utilization. Well-known high-tech solutions, such as gas turbine power stations (GTPS) based on an aircraft gas turbine engine [2–4], which generate electricity and thermal energy, have proven useful in the field of APG utilization. However, in the conditions of distributed generation [5,6], the use of autonomous power supply sources in parallel networking mode gives rise to the problem of reducing the operational characteristics of a GTPS based on gas turbine engines, such as the available electric power of gas turbine units (GTU) [7], energy efficiency [8], and especially power quality [9]. The decrease in GTPS performance can be explained by the new emerging properties of power supply systems (PSS), which create new conditions and put forward new requirements for the GTU operation.

Thus, the aim of the work was to study the influence of GTPS, based on gas turbine engines, on the quality of electrical energy in the electrical complex of the oil field. This will expand the set of acceptable solutions for the implementation of various methods and

control algorithms for automatic control systems (ACS), taking into account the territorial and geological conditions and factors of oil fields [10,11].

At the first stage of the research, a mathematical description of the electrical complex of an oil field was formed, taking into account the nonlinearity of a two-shaft gas turbine unit.

At the second stage of the research, instrumental measurements of the GTPS power consumption parameters (voltage levels, current loads, power consumption, quality parameters of electrical energy) were carried out in various operating modes. The analysis of the results of instrumental measurements made it possible to identify the GTPS operating modes in which an excess of the standardized values of the quality parameters of electrical energy is observed.

Basic recommendations have been developed to ensure the quality parameters of electrical energy.

At the final stage of the work, the analysis of the adequacy of the mathematical description of the electrical complex of the oil and gas field was carried out.

## 2. Mathematical Description of the Power Supply System

The calculating algorithm for the dynamic modes of the power supply system is multistage [12,13]. At first, the nodal voltages calculation [14] is performed using equations in a single generalized form of Equation (1). Then, the differential equations are solved, and the currents are found. The procedure is repeated at each step of the numerical integration of differential equations.

$$p\mathbf{I}_i = -\mathbf{A}_i\mathbf{U}_i \text{ - } \mathbf{B}_i\mathbf{I}_i - \mathbf{H}_i, \tag{1}$$

where $\mathbf{I}_i$ is a currents vector of element $i$, $p\mathbf{I}_i$ is a currents derivative vector of element $i$, $\mathbf{U}_i$ is the vector of voltages applied at the terminals of the element $i$, $\mathbf{A}_i$ and $\mathbf{B}_i$ are matrices whose dimension depends on the coordinate system in which the structural element is modeled, as well as on whether these equations are complete or simplified, and $\mathbf{H}_i$ is the vector of impact on the element $i$ [15].

The main models of elements in the transformed coordinates d and q of the PSS are shown (the generally accepted notations are used when writing systems of mathematical description):

- Main power system: $\mathbf{A} = \begin{pmatrix} 1/x_{PS} & 0 \\ 0 & 1/x_{PS} \end{pmatrix}$, $\mathbf{B} = \begin{pmatrix} 0 & -\omega \\ \omega & 0 \end{pmatrix}$, $\mathbf{H} = \begin{pmatrix} -E_d/x_{PS} \\ -E_q/x_{PS} \end{pmatrix}$;

- Synchronous generator (SG):

$$\mathbf{A} = \left[ \begin{pmatrix} x_d & 0 \\ 0 & x_d \end{pmatrix} - \begin{pmatrix} x_{ad} & x_{ad} & 0 \\ 0 & 0 & x_{aq} \end{pmatrix} \begin{pmatrix} x_f & x_{ad} & 0 \\ x_{ad} & x_D & 0 \\ 0 & 0 & x_Q \end{pmatrix}^{-1} \begin{pmatrix} x_{ad} & 0 \\ x_{ad} & 0 \\ 0 & x_{aq} \end{pmatrix} \right]^{-1},$$

$$\mathbf{B} = \mathbf{A} \times \begin{pmatrix} r & \omega x_q & -r_f \frac{x_D x_{ad} - x_{ad}^2}{x_f x_D - x_{ad}^2} & \omega x_{aq} \\ -\omega x_d & r & -\omega x_{ad} & -r_Q \frac{x_{aq}}{x_Q} \end{pmatrix}, \mathbf{H} = \mathbf{A} \times \begin{pmatrix} \frac{x_D x_{ad} - x_{ad}^2}{x_f x_D - x_{ad}^2} \\ 0 \end{pmatrix} U_f;$$

- Power transformer (T): $\mathbf{A} = -\begin{pmatrix} 1/x_T & 0 \\ 0 & 1/x_T \end{pmatrix}$, $\mathbf{B} = \begin{pmatrix} r_T/x_T & -\omega \\ \omega & r_T/x_T \end{pmatrix}$, $\mathbf{H} = 0$;

- Cable (CL) and overhead (OL) power lines (PL): $\mathbf{A} = -\begin{pmatrix} 1/x_{PL} & 0 \\ 0 & 1/x_{PL} \end{pmatrix}$,

$\mathbf{B} = \begin{pmatrix} r_{PL}/x_{PL} & -\omega \\ \omega & r_{PL}/x_{PL} \end{pmatrix}$, $\mathbf{H} = 0$;

- Static load (SL): $\mathbf{A} = -\begin{pmatrix} 1/x_{SL} & 0 \\ 0 & 1/x_{SL} \end{pmatrix}$, $\mathbf{B} = \begin{pmatrix} r_{SL}/x_{SL} & \omega \\ -\omega & r_{SL}/x_{SL} \end{pmatrix}$, $\mathbf{H} = 0$;

- Induction motor: $\mathbf{A} = -\left[ \begin{pmatrix} x_s & 0 \\ 0 & x_s \end{pmatrix} - \begin{pmatrix} x_m & 0 \\ 0 & x_m \end{pmatrix} \begin{pmatrix} x_r & 0 \\ 0 & x_r \end{pmatrix}^{-1} \begin{pmatrix} x_m & 0 \\ 0 & x_m \end{pmatrix} \right]^{-1}$,

  $\mathbf{B} = \mathbf{A} \times \begin{pmatrix} r & \omega x_s - \frac{\omega_2 x_m^2}{x_r} & -r_2 \frac{x_m}{x_r} & (\omega - \omega_2) x_m \\ \frac{\omega_2 x_m^2}{x_r} - \omega x_s & r & (\omega_2 - \omega) x_m & -\frac{r_2 x_m}{x_r} \end{pmatrix}$, $\omega_2 = (\omega - \omega_{IM})$, $\mathbf{H} = 0$.

It is advisable to use simplified identification models, which, however, should adequately reproduce the energy transients and the nonlinearity of the characteristics of GTU for the study of them as part of a complex model. The GTU interacts with the SG mechanically; thus, bringing the models of GTUs to a single generalized form (1) is not required [16,17].

The importance of taking into account the nonlinearity of the GTU model due to the presence of a system of several rotating masses is noted by many authors [16–20]. A nonlinear model of a GTU that takes into account the energy accumulation in the rotating masses of a two-shaft GTU [21,22] is presented as:

$$
\begin{cases}
\dot{A}_{DI} = \frac{(A_{DIZ} - A_{DI})}{T_{DI}}, \\
G_{TS} = f(A_{DI}), \\
\dot{G}_T = \frac{(\dot{G}_{TS} - G_T)}{T_{GT}}, \\
n_{TS} = f(G_T), \\
\dot{n}_{TK} = \frac{(n_{TS} - n_{TK})}{T_{NTK}}, \\
N_E = f(n_{TK}), \\
\Delta \dot{N}_C = \frac{K_N \frac{\dot{N}_E - N_G}{n_{CT}} - \Delta N_C}{T_N}, \\
\dot{n}_{CT} = \frac{\dot{N}_E - N_G}{T_{NCT} \cdot n_{CT}} + \Delta N_C,
\end{cases}
$$

where $A_{DI}$ is the angle of rotation of the gas dispenser, $A_{DIZ}$ is the specified rotation angle of the gas dispenser, $T_{DI}$ is the time constant of the fuel dispenser, $G_{TS}$ is fuel consumption by static characteristic, $G_T$ is fuel consumption, $T_{GT}$ is the time constant of fuel consumption, $n_{TS}$ is the speed of rotation of the turbine compressor rotor according to the static characteristic, $n_{TK}$ is the speed of rotation of the turbine compressor rotor, $T_{NTK}$ is the time constant of the turbine compressor rotor, $N_E$ is the available power of the free turbine (FT), $\Delta N_C$ is the value reflecting the effect of the derivative of the power imbalance on the FT rotor, $K_N$ is the gain of the change rate of the power imbalance, $N_G$ is the power consumption of the FT, $n_{CT}$ is the speed of rotation of the FT rotor, $T_N$ is the time constant of the influence of the change rate of the power imbalance on the FT rotation speed, and $T_{NCT}$ is the time constant of the FT rotor.

## 3. Model of Oil and Gas Field Power Supply System

The presented method of the model parameters calculating PSS elements allows simulating complex modes of PSS powered by local sources [23]. For example, Scheme 1 shows the typical power distribution scheme of a gas turbine power station, the main function of which is the associated petroleum gas utilization from the oil and gas field. During the operation of the GTPS, it is necessary to consider not only the operating conditions of power plants, depending on the fuel supply modes, but also the influence of bidirectional power flows from nearby substations (SS).

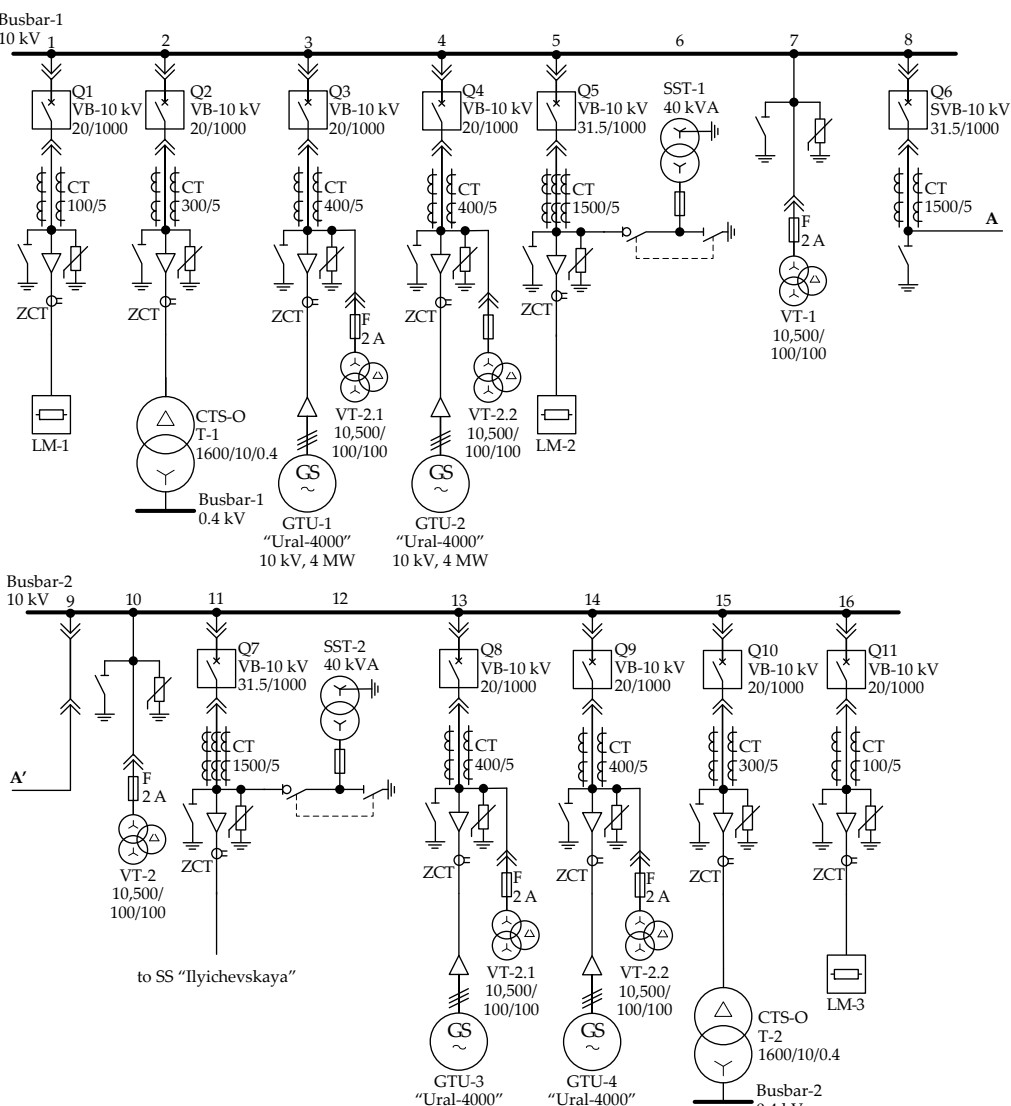

**Scheme 1.** The power distribution scheme of a gas turbine power station (where CT is a current measuring transformer, CTS-O is a complete transformer substation (outdoor design), LM is a load module, SST is a station service transformer, (S)VB is a (sectional) vacuum breaker, VT is a voltage measuring transformer, F is a fuse, ZCT is a zero-sequence current transformer).

The model of the oil and gas field's PSS was developed for the purpose of increasing the accuracy in modeling interrelated electromechanical and electromagnetic processes in the "GTPS-to-main-power system", which are not reflected in the project documentation.

The main power distribution unit of the GTPS is considered to be the complete modular switchgear (CSGM) 10 kV. Thus, the CSGM includes two busbar sections (BS). GTU-1 and GTU-2, as well as load modules (LM-1 and LM-2), are connected to BS-1 by switchgear units 3, 4, 1, and 5, accordingly. GTU-3, GTU-4, and LM-3 are connected to BS-2 by switchgear units 13, 14, and 16, accordingly. Connection with the main power system is also carried out from BS-2 (cell 11) by cable line 10 kV to the SS 35 kV "Ilyichevskaya" (1 × 4MVA). The GTU 4MW connection is carried out by an automatic synchronization block.

A sectional vacuum breaker (SVB) 10 kV (cell 8) is installed on the CSGM in order to ensure electrical connection between the 10 kV busbar sections for the distribution of electrical power to the main network from each of the GTUs.

In addition, the CTS-O with two power transformers of 1600 kVA is connected to the GTPS "Ilyichevskaya" (by cells 2 and 15), which includes the loads of the booster

compressor modules (BCM) and the low-voltage loads of the GTPS units themselves. Furthermore, the GTPS "Ilyichevskaya" GPP provides the power supply to the oilfield and nearby settlement loads connected to the 10 kV busbar of the SS 35 kV "Ilyichevskaya" (Scheme 2) [15].

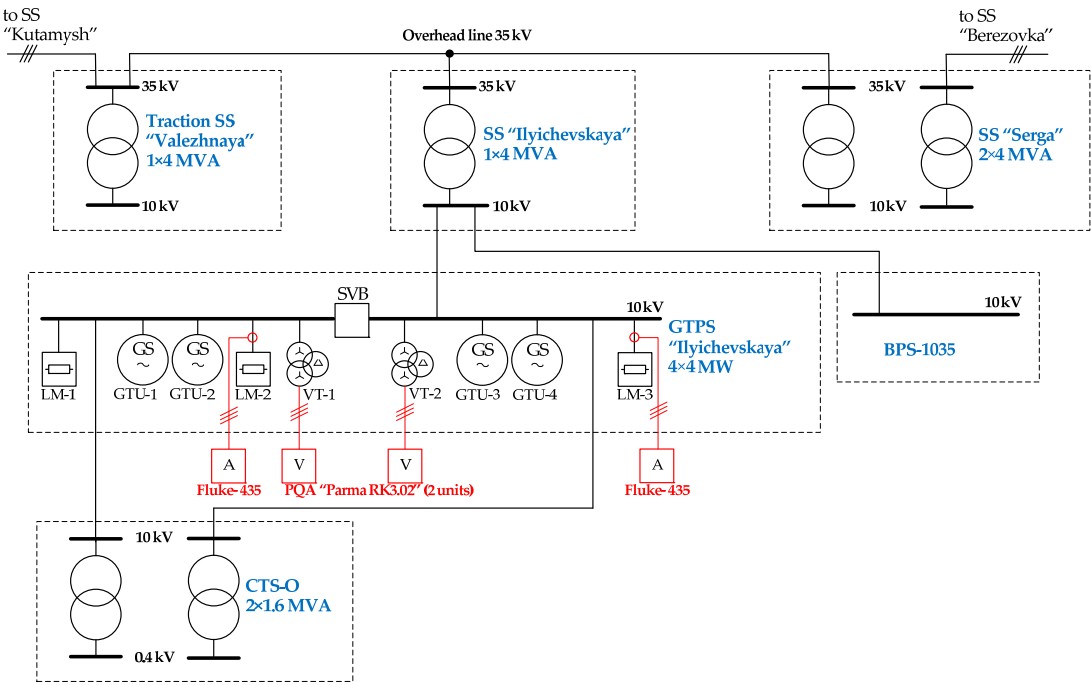

**Scheme 2.** One-line diagram of the GTPS "Ilyichevskaya" (with the connection diagram of measuring instruments for analyzing the parameters of the PSS, where PQA is power quality analyzer, BPS is booster pump station).

The power distribution scheme of the GTPS "Ilyichevskaya" provides 10 main operating modes, depending on the amount of APG burned, the repaired elements count, and the amount of electric energy generated in the external network. The modes under consideration include the operation of the GTUs of the GTPS "Ilyichevskaya" both in local mode and in parallel with the main power system.

The SVB (cell 8) and the main network connection unit (cell 11) are not equipped with automatic synchronization blocks; therefore, switching to the parallel operation mode requires turning on these cells before turning on the GTU cells (3, 4, 13, and 14).

## 4. Experiment Planning

The instrumental survey of the power quality indicators (PQI) was carried out at the following nodes of the oil and gas field's PSS:

1. Busbar-1 of the GTPS;
2. Busbar-2 of the GTPS.

The power quality analyzers were connected to the secondary measuring circuits of the VTs (cells 7 and 10) with a transformation coefficient $k_T$ = 105. The currents on the 0.4 kV busbars of the power transformer, including the LM-2 and LM-3 of the GTPS "Ilyichevskaya", were measured in order to identify the influence of the load on the PQI [9] (during the instrumental survey, the LM-1 was taken out for repair, so no measurements were carried out on it).

## 5. Measurement Tools and Methods

As a means of measuring the power quality, the recorders of power quality "Parma RK3.02" [24] were used. These devices comply with the requirements of IEC 61000-4-30:2008

"Electric energy. Electromagnetic compatibility of technical equipment. Power quality measurement methods" [25] and CSA IEC 61000-4-7-2017 "Electromagnetic compatibility (EMC)" [26].

The current was measured using power quality analyzers FLUKE-435 [27]. The devices were connected by flexible current clamps with a transformation coefficient of 3000:1, and for voltage measuring clamps with a transformation coefficient of 1:1.

Taking into account the requirements of Russian Standard GOST R 32144-2013 "Electric energy. Electromagnetic compatibility of technical equipment. Power quality limits in the public power supply systems" [28] (complies with the European regional standard EN 50160:2010 "Voltage characteristics of electricity supplied by public distribution network" [29]), compliance with the standards of the following voltage PQI were investigated:

- Frequency deviation;
- Slow voltage changes (over-deviation and under-deviation);
- Coefficient of voltage unbalance by the zero and reverse sequence;
- Voltage total harmonic distortion (THD);
- Coefficient of the 2nd to 40th harmonic components of the voltage;
- Coefficient and duration of voltage swell;
- Depth and duration of the voltage dip.

Furthermore, according to the requirements of standards [28,29], the length of the time interval of measurements was assumed to be seven days, and the total number of measured intervals at the measuring points was five. The number of measured intervals is determined by the need to cover all operating modes of the GTPS.

## 6. Experimental Results

As a result, there was an overstepping in the normally permissible limits of the 11th and 13th harmonic components; however, it was within the permissible limits. According to the Operational Log, the transfer from the local operation mode of the GTPS "Ilyichevskaya" to parallel operation with the main power system was carried out during this period (27 July from 2:30 pm to 3:35 pm). The switching moment is shown in Figure 1.

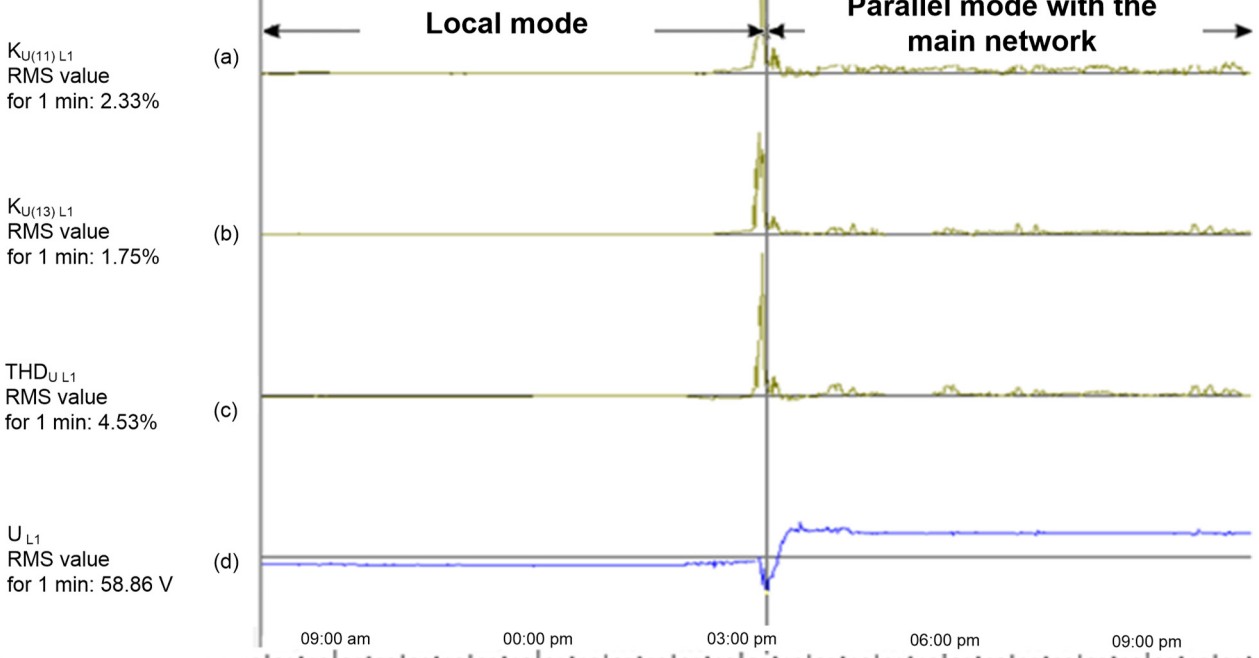

**Figure 1.** The PQI profile from 27 July 9:00 am to 27 July 10:00 pm: (**a**) coefficient of the 11th harmonic component, (**b**) coefficient of the 13th harmonic component, (**c**) THD, and (**d**) phase voltage in the secondary measuring circuit.

It can be seen that mode-to-mode switching increases in the level of harmonic components. In local mode, the harmonic components in the measurement nodes are practically absent (the values of the harmonic component coefficient are less than 0.2%). It can be seen that after switching from the local mode of the GTPS "Ilyichevskaya" to the parallel mode with the main network, the coefficient of harmonic components increases significantly (up to 1.5%). We can conclude that the source of a short-term overstepping of the normally permissible limits of the harmonic components is modes switching, and the source of long-term fixed disturbances is located in the main power system (cell 11 in Scheme 1).

During the PQI registration at the GTPS "Ilyichevskaya", voltage dips and swells were recorded. Depth and duration of the voltage dips are shown in Tables 1 and 2. Values and duration of the voltage swells are shown in Table 3.

**Table 1.** Voltage dip registration results at busbar-2.

| Parameter | From 27 July to 28 July | | |
| | L1 | L2 | L3 |
| --- | --- | --- | --- |
| Quantity | 5 | 7 | 5 |
| -//- sum | | 17 | |
| Maximum depth, % | 89.38 | 89.86 | 89.24 |
| Maximum duration, ms | 20 | 10 | 20 |
| Parameter | From 28 July to 29 July | | |
| | L1 | L2 | L3 |
| Quantity | 1 | 2 | 1 |
| -//- sum | | 4 | |
| Maximum depth, % | 72.83 | 84.69 | 85.31 |
| Maximum duration, ms | 660 | 1250 | 350 |

**Table 2.** Voltage dip registration results at busbar-1.

| Parameter | From 30 July to 31 July | | |
| | L1 | L2 | L3 |
| --- | --- | --- | --- |
| Quantity | 1 | 0 | 0 |
| -//- sum | | 1 | |
| Maximum depth, % | 87.59 | 0.00 | 0.00 |
| Maximum duration, ms | 10 | 0 | 0 |
| Parameter | From 2 August to 3 August | | |
| | L1 | L2 | L3 |
| Quantity | 2 | 1 | 2 |
| -//- sum | | 5 | |
| Maximum depth, % | 83.59 | 88.41 | 89.66 |
| Maximum duration, ms | 20 | 10 | 10 |

**Table 3.** Voltage swell registration results at busbar-1.

| Parameter | From 2 August to 3 August | | |
| | L1 | L2 | L3 |
| --- | --- | --- | --- |
| Quantity | 0 | 2 | 0 |
| -//- sum | | 2 | |
| Maximum depth, % | 0.00 | 1.35 | 0.00 |
| Maximum duration, ms | 0 | 90 | 0 |

Measurement of current loads and power flows by the FLUKE-435 (connected to the 0.4 kV busbar of the LM-3) was performed in the period from 22 July 11:00 pm to 25 July 4:00 am with a sampling rate of 0.5 s. During the period under review, current changes from 1295 to 2570 A were recorded at the moments of operation of the LM-3. It should be

noted that the current consumption of the LM-3 has some unbalance, which causes current in the neutral conductor (up to 51 A). In addition, current fluctuations in individual phases were recorded (a decrease in current in L1 at 02:10 on 25 July by approximately 20 A for half an hour), which, due to the absence of voltage changes at a given time, may be due to switching of individual resistive elements of the LMs.

During the PQI registration at busbar-2 of the GTPS "Ilyichevskaya", inconsistencies of the voltage THD were recorded (Table 4).

Registered excess of 11th, 13th, 17th, 19th, 21st, 23rd, 25th, 27th, 29th, and 31st harmonic component coefficient of the voltage was observed on 31 July from 17:00 to 18:00 (Figure 2). During this period, according to the Operational Log, an emergency shutdown of the power units of the GTPS "Ilyichevskaya" was recorded, followed by a return to the mode of parallel operation with the main power system. The characteristic sources of harmonic components in the PSS were:

- Launching of BCM-1, BCM-3 (busbar-1 of the CTS-O 2×1.6 MWA), and BCM-4 (busbar-2 of the CTS-O 2×1.6 MWA) from the soft start device (SSD) for orders 11th, 13th;
- Switching voltage dips for the 23rd, 25th, 27th, 29th, 31st harmonic component coefficient.

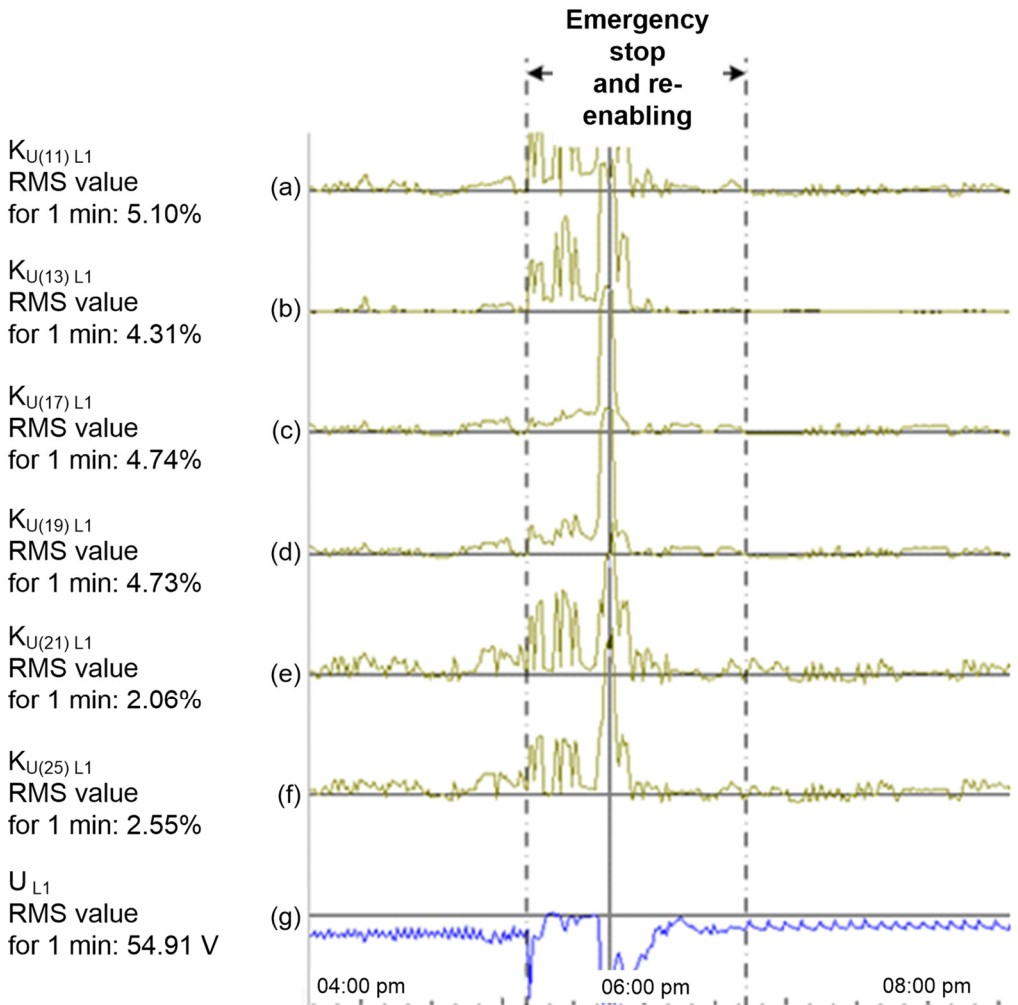

**Figure 2.** The PQI profile 31 July from 4:00 pm to 9:00 pm. (**a**) Coefficient of the 11th harmonic component, (**b**) coefficient of the 13th harmonic component, (**c**) coefficient of the 17th harmonic component, (**d**) coefficient of the 19th harmonic component, (**e**) coefficient of the 21st harmonic component, (**f**) coefficient of the 25th harmonic component and (**g**) phase voltage in the secondary measuring circuit.

**Table 4.** Voltage THD% values (from 30 July to 31 July).

| Phase | 95% of Time Range | | | | 100% of Time Range | | | |
|---|---|---|---|---|---|---|---|---|
| | The Normally Permissible Limits, THD$_{norm}$ | The Measurement Results | | Conclusion | The Maximum Permissible Limits, THD$_{max}$ | The Measurement Results | | Conclusion |
| | | THD% | Duration | | | THD% | Duration | |
| L1 | 5.00 | 1.85 | 0.53 | valid | 8.00 | 12.57 | 0.38 | invalid |
| L2 | 5.00 | 1.79 | 0.54 | valid | 8.00 | 12.43 | 0.38 | invalid |
| L3 | 5.00 | 1.81 | 0.53 | valid | 8.00 | 12.54 | 0.38 | invalid |

Based on the absence of characteristic sources of the 17th, 19th, and 21st harmonic components in the PSS of the GTPS "Ilyichevskaya", it follows that the source of these interferences is located in the main power system (cell 11 of the CSGM-10 kV). The orders of harmonic components are determined by the operation of 6-pulse bridge converters, including frequency converters with a DC link.

## 7. Analysis of the Influence of GTPS Operating Modes on the Power Quality

In order to determine the possible influence of the used mode on the PQI and to separate the disturbances into local (from its own generation and consumers) and main power system, a joint analysis of the generation modes of the GTPS "Ilyichevskaya" and the detected deviations of the PQI values was carried out.

The map of the main operating modes used by the GTPS "Ilyichevskaya" is summarized in Table 5.

**Table 5.** Switching breaker status of main operating modes GTPS "Ilyichevskaya".

| Mode | Cell 1 | Cell 3 | Cell 4 | Cell 5 | Cell 8 | Cell 11 | Cell 13 | Cell 14 | Cell 16 |
|---|---|---|---|---|---|---|---|---|---|
| | LM-1 | GTU-1 | GTU-2 | LM-2 | SVB | to SS "Ilyichevskaya" | GTU-3 | GTU-4 | LM-3 |
| no. 1 | ON | ON | ON | ON | ON | ON | ON | ON | ON |
| no. 2 | ON | ON | ON | ON | off | ON | ON | ON | ON |
| no. 3 | off | off | ON | ON | off | ON | ON | ON | ON |
| no. 4 | ON | ON | off | off | off | ON | ON | ON | ON |
| no. 5 | ON | ON | ON | ON | off | ON | off | ON | off |
| no. 6 | ON | ON | ON | ON | off | ON | ON | off | off |
| no. 7 | off | off | off | off | off | ON | ON | ON | off |
| no. 8 | off | off | off | off | off | ON | ON | ON | ON |
| no. 9 | ON | ON | ON | ON | off | ON | off | off | off |
| no. 10 | ON | ON | ON | ON | off | off | ON | ON | ON |

Not always taking into account repair switches and schemes, the output of power is carried out according to one of the design modes. Thus, it is necessary to correlate the status of the CSGM-10 kV switching breakers during the measurements (Table 6) with the map of the main operating modes.

The following looks at the main operating modes from Table 5 in more detail.

**Table 6.** Map of the operating switches at the GTPS "Ilyichevskaya" from 17 June to 10 August.

| Date | Time | Cell 1 | Cell 3 | Cell 4 | Cell 5 | Cell 8 | Cell 11 | Cell 13 | Cell 14 | Cell 16 |
|---|---|---|---|---|---|---|---|---|---|---|
| | | LM-1 | GTU-1 | GTU-2 | LM-2 | SVB | to SS "Ilyichevskaya" | GTU-3 | GTU-4 | LM-3 |
| 17 June | 15:00 | ON | ON | off | off | ON | ON | ON | ON | ON |
| 19 June | 14:45 | off | ON | off | ON | ON | ON | ON | ON | ON |
| 20 June | 8:15 | off | ON | off | off | ON | ON | ON | ON | ON |
| 20 June | 8:40 | off | ON | off | ON | ON | ON | ON | ON | ON |
| 20 June | 13:10 | off | ON | off | off | ON | ON | ON | ON | ON |
| 20 June | 17:00 | off | ON | off | ON | ON | ON | ON | ON | ON |
| 20 June | 17:25 | off | off | off | off | ON | ON | off | off | off |
| 20 June | 21:00 | ON | ON | off | ON | ON | ON | ON | ON | off |
| 21 June | 9:20 | off | off | off | off | ON | ON | off | off | off |
| 21 June | 12:50 | off | ON | off | ON | ON | ON | ON | ON | ON |
| 22 June | 12:45 | ON | ON | off | ON | ON | ON | ON | ON | off |
| 24 June | 20:25 | off | off | off | off | ON | off | off | off | off |
| 25 June | 0:10 | off | ON | off | off | ON | off | ON | ON | off |
| 25 June | 1:50 | ON | ON | off | ON | ON | ON | ON | ON | off |
| 25 June | 10:10 | ON | ON | off | off | ON | ON | ON | off | off |
| 25 June | 10:30 | ON | ON | off | off | ON | off | ON | off | off |
| 5 July | 1:00 | off | off | off | off | off | off | off | off | off |
| 13 July | 22:00 | ON | ON | off | ON | ON | ON | off | ON | off |
| 14 July | 23:00 | ON | ON | ON | ON | ON | ON | off | ON | off |
| 16 July | 12:35 | ON | ON | ON | ON | ON | ON | off | ON | ON |
| 16 July | 17:40 | off | ON | ON | ON | ON | ON | off | ON | off |
| 16 July | 18:30 | ON | ON | ON | ON | ON | ON | off | ON | ON |
| 17 July | 8:20 | off | ON | off | off | ON | ON | off | off | off |
| 17 July | 9:40 | ON | ON | ON | ON | ON | ON | off | ON | ON |
| 18 July | 12:00 | off | ON | ON | off | ON | ON | off | ON | ON |
| 18 July | 15:00 | ON | ON | ON | ON | ON | ON | off | ON | ON |
| 18 July | 18:40 | ON | ON | off | ON | ON | ON | ON | ON | ON |
| 22 July | 11:05 | off | off | off | off | ON | ON | off | off | off |
| 22 July | 16:05 | ON | ON | off | ON | ON | ON | ON | ON | ON |
| 23 July | 19:40 | ON | ON | off | ON | ON | off | ON | ON | ON |
| 26 July | 8:40 | ON | ON | off | ON | ON | off | off | ON | ON |
| 26 July | 18:35 | ON | ON | off | ON | ON | off | ON | ON | ON |
| 27 July | 16:35 | ON | ON | off | ON | ON | ON | ON | ON | ON |
| 29 July | 14:30 | off | off | off | off | ON | ON | off | ON | off |
| 29 July | 16:00 | ON | ON | off | off | ON | ON | ON | ON | ON |
| 1 August | 10:50 | ON | ON | off | off | ON | ON | ON | off | ON |
| 3 August | 22:20 | ON | ON | off | off | ON | ON | ON | ON | ON |
| 7 August | 16:55 | ON | ON | ON | off | ON | ON | off | ON | ON |

### 7.1. Operating Mode No.1

The GTU-1, GTU-2, GTU-3, GTU-4, and LM-1, LM-2, LM-3 are in operation. Two busbars of CSGM-10 kV and 10 kV busbar of the SS 35 kV "Ilyichevskaya" are switched on parallel operation.

In this mode, it is possible to supply electricity to the main power system and (or) ensure uninterrupted power supply to oil and gas field loads connected to the 10 kV busbar of the SS 35 kV "Ilyichevskaya".

The most similar modes were observed (Table 6):

(a) From 16 July 00:35 pm to 17 July 8:20 am (the GTU-3 (cell 13) is disconnected). In the considered time interval, the PQI did not exceed the normally permissible limits (Figure 3). According to the Operational Log, from 17:40 to 18:30 on 16 July, there was an exit from the operating mode due to a spontaneous power reset by the LM-3 (2 MW). During this emergency disturbance, the PQI did not go beyond the permissible values, except for the frequency deviation (which is within the permissible time limits), which indicates the high stability of this mode.

(b) From 04:35 pm on 27 July to 02:30 pm on 29 July (the GTU-2 (cell 4) is disconnected). In the considered time interval, the PQI did not exceed the normally permissible limits (Figure 4). Small voltage fluctuations with a duration of less than 5 min, observed during the operation mode, can be caused by the BCM-4 control system operation from 11:50 am to 06:30 pm.

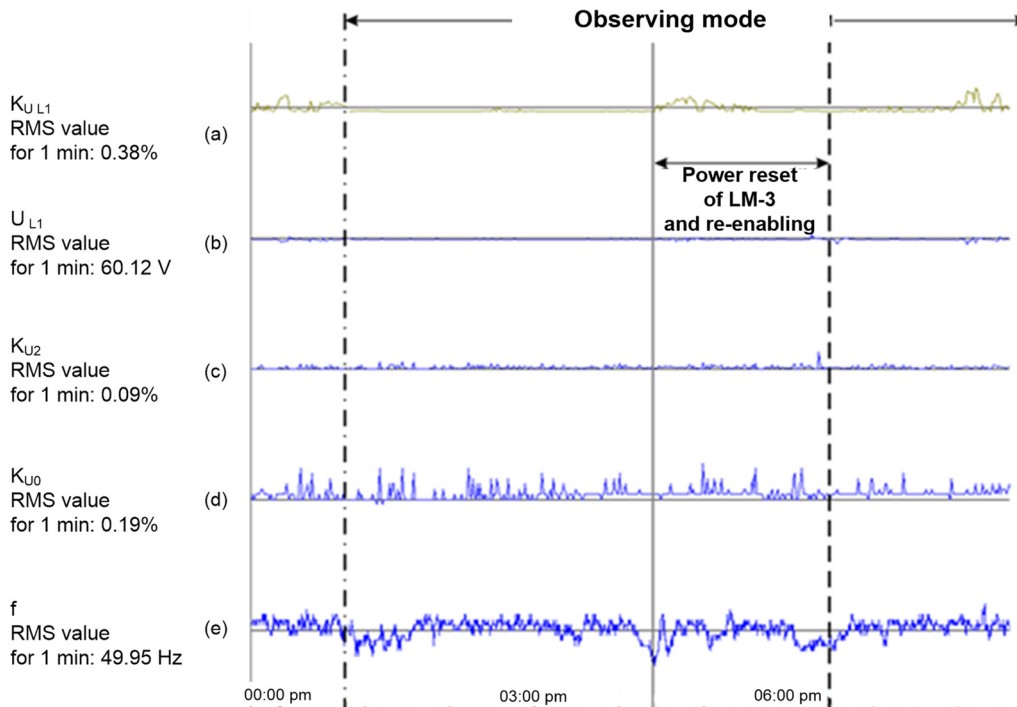

**Figure 3.** The PQI profile on 16 July from 11:40 am to 8:30 pm. (**a**) Non-sinusoidal coefficient, (**b**) phase voltage in the secondary measuring circuit, (**c**) negative and (**d**) zero-sequence voltage unbalance and (**e**) frequency.

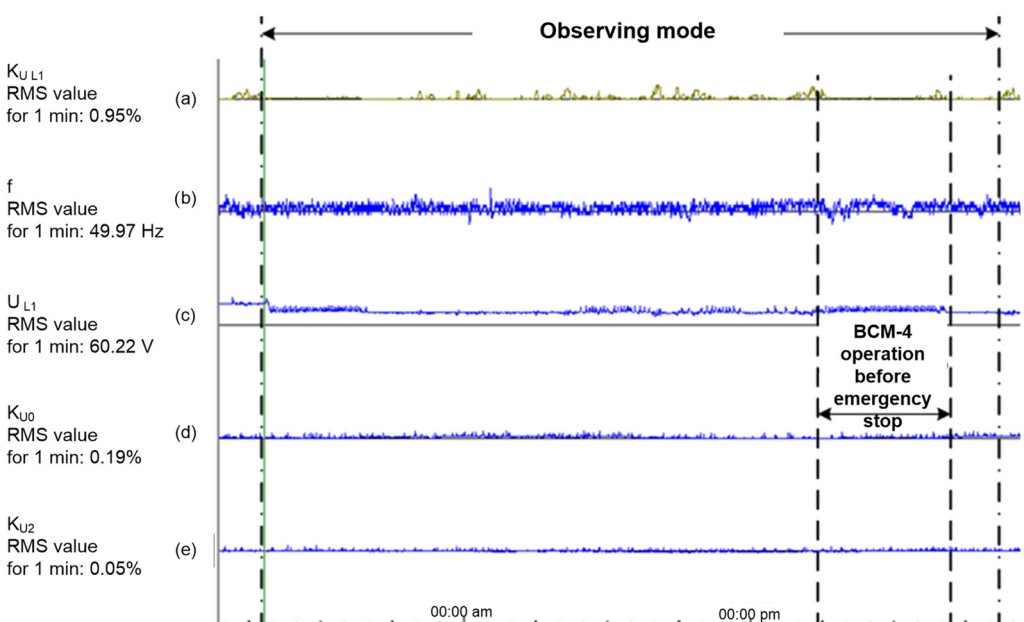

**Figure 4.** The PQI profile on 28 July from 11:40 am to 8:30 pm. (**a**) Non-sinusoidal coefficient, (**b**) frequency, (**c**) phase voltage in the secondary measuring circuit, (**d**) the zero and (**e**) negative-sequence voltage unbalance.

### 7.2. Operating Mode No. 2

The GTU-1, GTU-2, GTU-3, GTU-4, and LM-1, LM-2, LM-3 are in operation. The SVB of the CSGM-10 kV is disconnected (busbar-1 is in local mode). The busbar-2 of CSGM-10 kV and 10 kV busbar of the SS 35 kV "Ilyichevskaya" are switched on parallel operation.

In this mode, it is possible to supply electricity to the main power system and (or) ensure an uninterrupted power supply to oil and gas field loads connected to the 10 kV busbar of the SS 35 kV "Ilyichevskaya".

Due to the absence of recorded shutdowns of the SVB (except for normal and emergency stop of the GTPS "Ilyichevskaya"), the mode under consideration was not observed during the period of instrumental measuring, so the effect of the mode on the busbar-2 is similar to the operation mode no. 1, which was observed from 10:30 am on 25 June to 01:01 am on 5 July (when the switch breaker is turned off on the 10 kV busbar of the SS 35 kV "Ilyichevskaya", and the GTU-1, GTU-3 and LM-1 are in operation). The analysis of this operation mode (Figure 5) revealed a certain spectrum of harmonic components, but within the normally permissible limits (up to 1% of the main harmonic), which, together with the idling characteristic of power units, can confirm the receipt of these harmonic components from the load, and not from the electrical power source. The zero and negative sequence voltage unbalance does not exceed 0.3% for the entire duration of the mode.

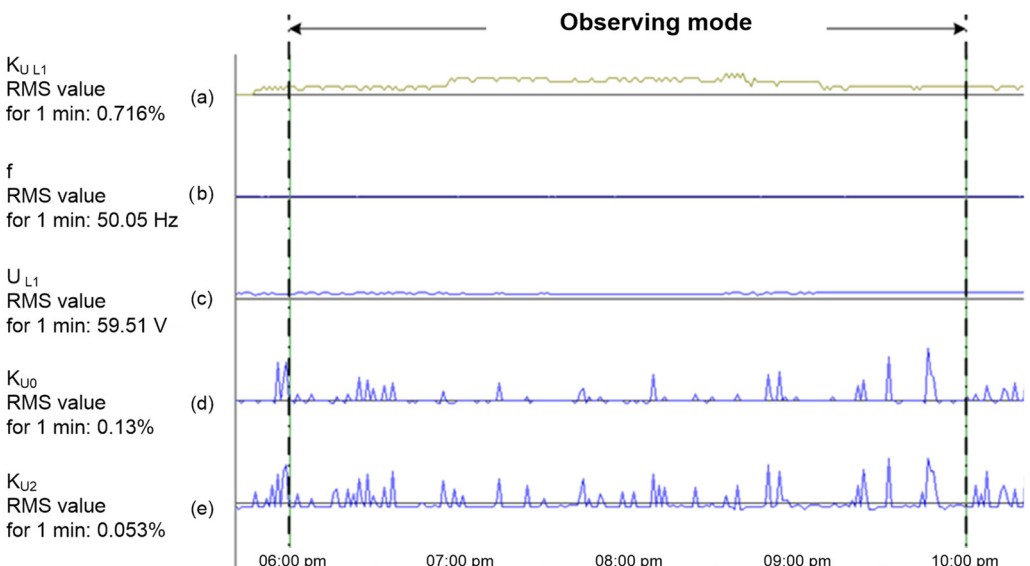

**Figure 5.** The PQI profile on 26 June from 06:00 am to 10:00 pm. (**a**) Non-sinusoidal coefficient, (**b**) frequency, (**c**) phase voltage in the secondary measuring circuit, (**d**) zero and (**e**) negative sequence voltage unbalance.

### 7.3. Operating Modes No. 3 and 4

The operating modes no. 3 and 4 are built on the basis of operating mode no. 2 by disconnecting one GTPS unit and one LM from busbar-1 of the CSGM-10 kV. In operating mode no. 3, LM-1 and GTU-1 are disconnected, and in operating mode no. 4, LM-2 and GTU-2 are disconnected, compared to mode no. 2. They are used when reducing the APG pressure on the flare and putting the GTU-1 and GTU-3 into repair/reserve. In these modes, it is possible to supply electricity to the main power system.

Figure 5 shows the profile of the measured PQI, which characterizes the operation in the closest mode to operating modes no. 3 and 4. The considered mode was observed from 00:50 pm on 21 June to 00:45 pm on 22 June (GTU-1, GTU-3, GTU-4, and LM-2 and LM-3 were in operation, the SVB is turned on). The parameters of this mode are close to the parameters of the mode shown in Figure 6 (operating mode no. 2), but there are more pronounced voltage ripples, which can be caused not only by switching oilfield loads, but also by regulating the power of the LMs. Despite short-term increases in the PQI, the excess of the normally permissible limits was not recorded.

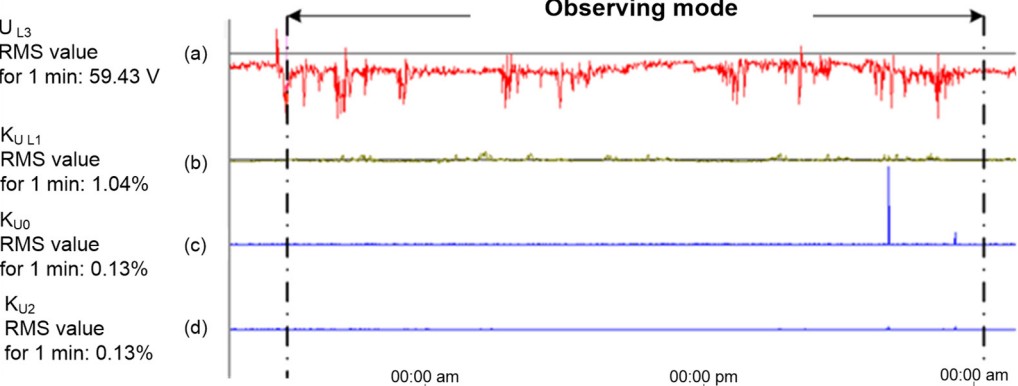

**Figure 6.** The PQI profile from 00:50 pm on 21 June to 00:45 pm on 22 June. (**a**) Phase voltage in the secondary measuring circuit, (**b**) non-sinusoidal coefficient, (**c**) zero and (**d**) negative sequence voltage unbalance.

### 7.4. Operating Modes No. 5 and 6

GTU-1, GTU-2, and LM-1 and LM-2 are connected to busbar-1 of the CSGM-10 kV in operating modes no. 5 and 6. The SVB is disabled. Busbar-2 in operating mode no. 5 is connected to the GTU-4, and in operating mode no. 6—GTU-3. The LM-3 is disabled in both modes; the power output from the busbar-2 is carried out on the CTS-O and the 10 kV bus section of the SS 35 kV "Ilyichevskaya". In these modes, it is possible to supply electricity to the main power system.

The operating mode of the GTPS "Ilyichevskaya", observed from 11:00 pm on 14 July to 00:35 pm on 16 July, is comparable to the operating modes no. 5 and 6 (GTU-1, GTU-2, GTU-4, LM-1, and LM-2 are in operation). Figure 7 shows the PQI profile that characterizes the operation mode under consideration. There were no deviations of the PQI beyond the normally permissible limits in this period. A smooth change in the zero-sequence voltage unbalance may be due to the regulation of the production of the GTPS "Ilyichevskaya" in the mode of maintaining the pressure on the APG flare (in this case, an increase in power when an excess of APG appears).

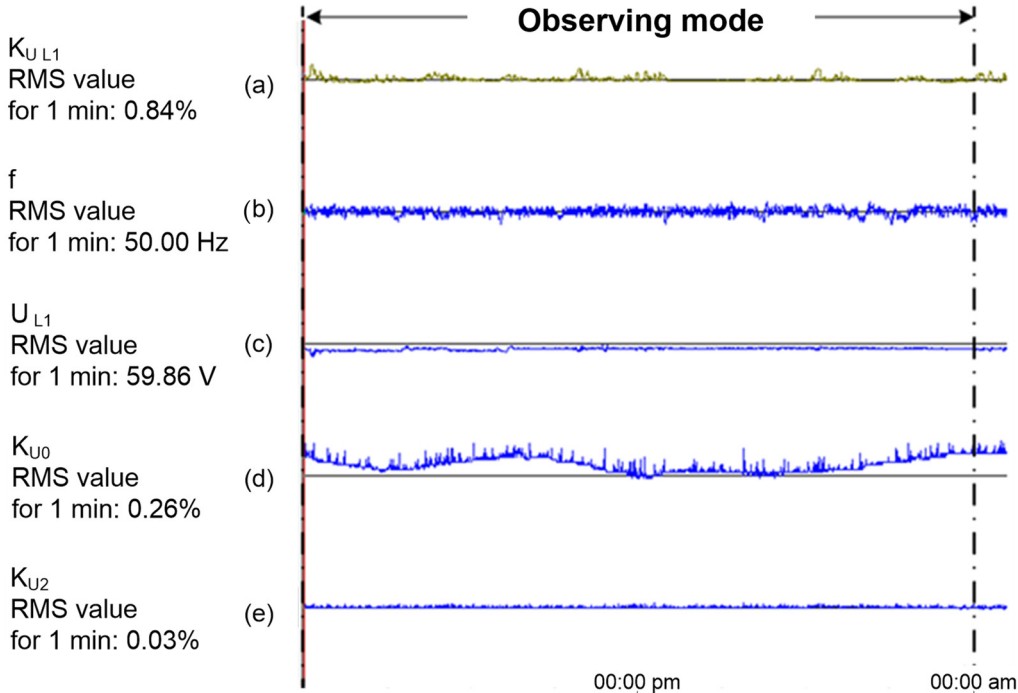

**Figure 7.** The PQI profile from 11:00 pm on 14 July to 00:35 pm on 16 July. (**a**) Non-sinusoidal coefficient, (**b**) frequency, (**c**) phase voltage in the secondary measuring circuit, (**d**) zero and (**e**) negative sequence voltage unbalance.

### 7.5. Operating Mode No. 7

In operating mode no. 7, all GTUs and LMs of the busbar-1 are disabled, GTU-3 and GTU-4 on busbar-2 are in operation. The power output from busbar-2 is carried out on the CTS-O and the 10 kV bus section of the SS 35 kV "Ilyichevskaya".

The period under consideration corresponds from 02:30 to 29 July 04:00 on 29 July (Figure 8). This mode is forced, and the transition to it was carried out by means of an emergency stop of the GTU-1 and GTU-3 (the transformer of the LM-2 burned down). The power unit of the GTPS no. 4, which operates on the external network, remains in operation. The PQI in this mode does not exceed the normally permissible limits.

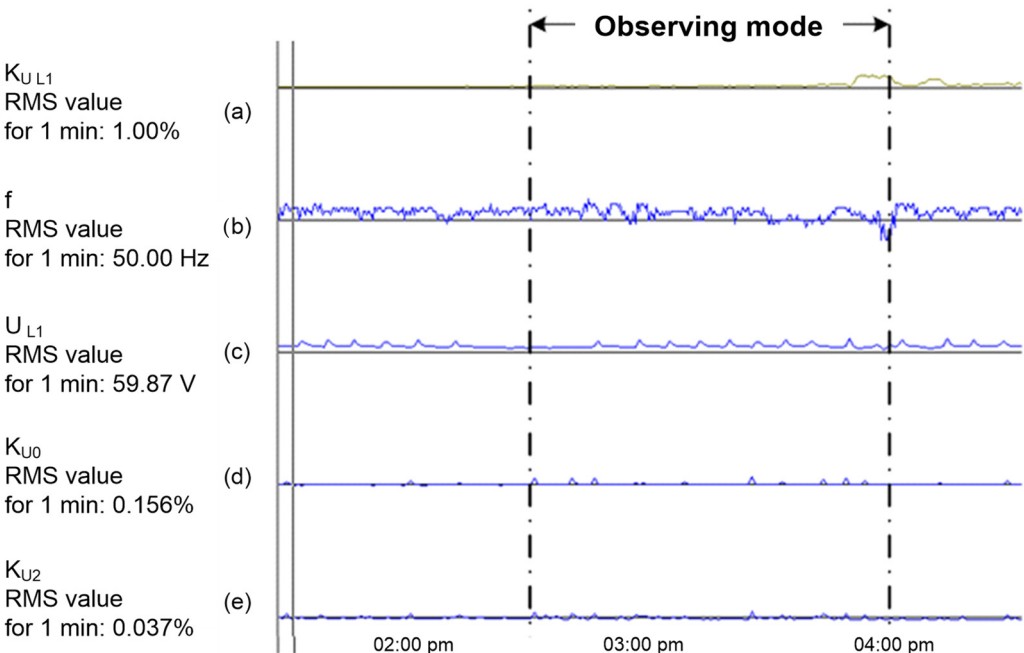

**Figure 8.** The PQI profile on 29 July from 02:30 pm to 04:00 pm. (**a**) Non-sinusoidal coefficient, (**b**) frequency, (**c**) phase voltage in the secondary measuring circuit, (**d**) zero and (**e**) negative sequence voltage unbalance.

### 7.6. Operating Mode No. 8

In operating mode no. 8, the busbar-1 is disabled, the GTU-3, GTU-4, and LM-3 on busbar-2 are in operation. The power output from busbar-2 is carried out on the LM-3 (cell 16), the CTS-O (cell 15), and the 10 kV bus section of the SS 35 kV "Ilyichevskaya" (cell 11).

A similar mode with the connected GTU-1 (cell 3) and SVB (cell 8) was observed from 08:40 am on 15 July to 06:35 pm on 16 July (Figure 9). The PQI in this mode did not exceed the normally permissible limits. Throughout the entire mode, stable operation is observed with small frequency fluctuations.

### 7.7. Operating Mode No. 9

In operating mode no. 9, the GTU-1, GTU-2, LM-1, and LM-2 on busbar-1 are in operation. The SVB (cell 8) is disconnected, but busbar-2 is energized (cell 11 is turned on) to power the CTS-O from the SS 35 kV "Ilyichevskaya".

In this mode, busbar-1 operates in a local mode, similar to busbar-1 in operating mode no. 2 (Figure 5), and the generation of power units of the GTPS "Ilyichevskaya" at busbar-2 is absent. The operating mode of busbar-2 cannot be compared with any of the recorded modes during the measuring, and the PQI completely depends on the parameters of the operating mode of the main power system.

### 7.8. Operating Mode No. 10

In operating mode no. 10, all the GTUs and LMs are in operation. The SVB (cell 8) and the main power system (cell 11) are disconnected. Both busbars operate separately.

In this mode, busbar-1 and busbar-2 work with parameters similar to those of busbar-1 in operating mode no. 2; therefore, the effect of operating mode no. 10 on the PQI is similar (Figure 5).

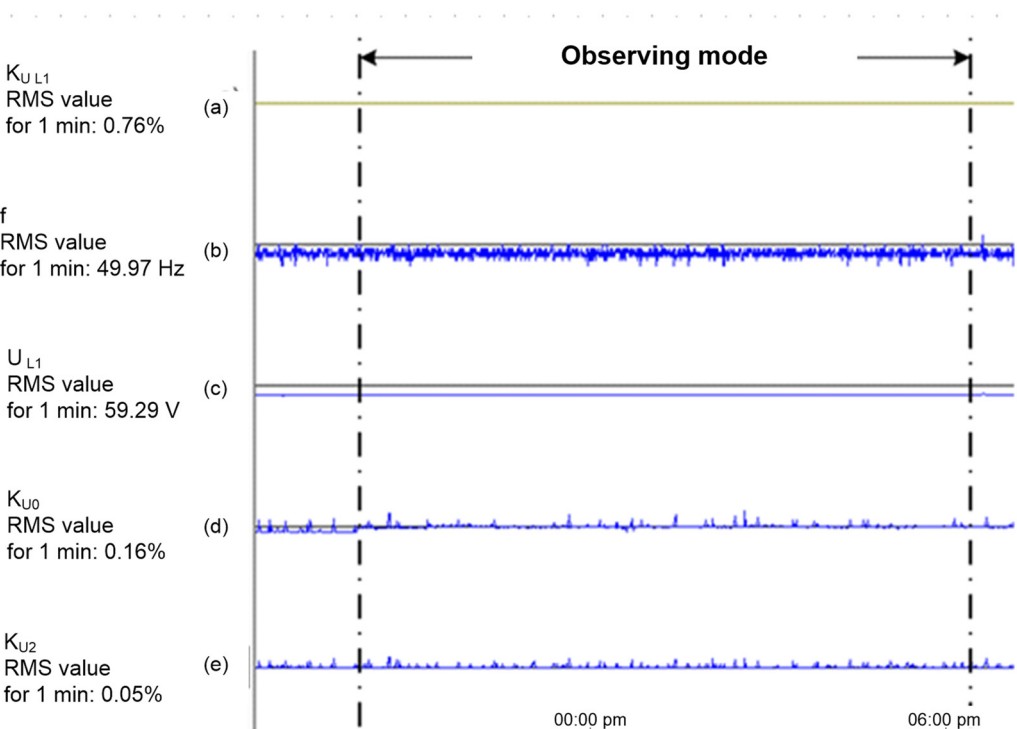

**Figure 9.** The PQI profile from 00:00 am on 15 July to 00:00 am on 16 July. (**a**) Non-sinusoidal coefficient, (**b**) frequency, (**c**) phase voltage in the secondary measuring circuit, (**d**) zero and (**e**) negative sequence voltage unbalance.

## 8. Discussion

No overflows of active power exceeding the normally permissible parameters of the equipment were detected during the measuring. However, spontaneous partial load discharges and emergency shutdowns of the LMs were recorded on 20 June, 16 July, 18 July, 29 July, leading to an emergency stop of the GTPS "Ilyichevskaya" power units. It is known that the average time for the resumption of the operating mode of the GTPS "Ilyichevskaya" is from 2 to 4 h, which causes an increase in emissions into the atmosphere with incomplete APG combustion on the flare.

The analysis of the spectrum and levels of harmonic components showed that the recorded excess of the permissible values of the coefficients of odd harmonics from 11th to 17th is due to the presence of powerful converter equipment in the load, in particular, frequency converters [30,31].

Furthermore, during the PQI measurements in the PSS of the GTPS "Ilyichevskaya", voltage dips and swells were registered. The maximum recorded voltage dip duration was 1250 ms, and the dip depth was 84.69% of the nominal value. The analysis of the registered dips and swells, as well as the absence of exceeding the permissible values of slow voltage changes, allows us to conclude that there are no uncontrolled overflows of reactive power along the main voltage harmonic during the period of the PQI measuring [32].

Analysis of the spectrum of harmonic components that go beyond the normally permissible values by the standard [28] and their distribution [33–35] allows us to conclude that the characteristic sources of the recorded deviations of the nth harmonic component coefficient in the PSS of the GTPS "Ilyichevskaya" are:

- Soft-start devices BKM-1,2, 3, 4, 5 for 5th, 7th, 11th, and 13th harmonics, only during the start of the drive motor;
- Operation of the pulse-phase control system with non-sinusoidality of the supply voltage for 6th harmonic;
- Operation in the main power system of 6-pulse bridge converters for 11th, 13th, 17th, 19th, and 21st harmonics;

- Switching of vacuum breakers and resonant phenomena for short-term harmonic components of odd order 23 and higher.

The reason for the recorded negative sequence voltage unbalance was a breakdown of the inter-turn insulation of the power transformer of the SS 35 kV "Ilyichevskaya", detected on 25 June and its subsequent asymmetric loading, since, after the repair work, no excess of the unbalance coefficient in the negative sequence of the normally permissible values was detected.

The reason for the recorded voltage dips and swells in the PSS was unfavorable weather conditions (lightning discharges), as well as switching of vacuum breakers [36] during an emergency stop of one or more power units of the GTPS "Ilyichevskaya".

In addition, an analysis of the influence of the operating modes of the GTPS "Ilyichevskaya" on the PQI was carried out, as a result of which the following was revealed:

- When the load is dropped from 1.5 MW, even when working in parallel with the main power system, there is an emergency stop of the power units of the GTPS "Ilyichevskaya".
- When BCM connected to the busbar, working in the local mode of generation, there was a slight (1%) fluctuation in the steady-state value of the voltage with a period of about 5 min.
- More frequent fluctuations in the level of voltage in view of the measured current loads of the LM-3 indicate frequent assist switching devices of the LM's switchgear 0.4 kV that can be associated with the misalignment of the control systems of the GTUs and LMs.
- The connection of LMs when entering the mode causes the frequency to decrease almost to the minimum permissible value in the presence of two GTPs or less in operation.
- Parallel operation of the GTPS "Ilyichevskaya" with the main power system is characterized by a higher THD compared to the local operation; however, during local operation, a wider range of fluctuations in the frequency of the voltage is observed.

The results of the comparative analysis of the results of the measured and model values (according to Equation (1) [12,15]) of the voltage and current of the GTPS "Ilyichevskaya" are shown in Table 7. The relative error of the calculation δ was estimated by the equation:

$$\delta = 100\frac{|x - X|}{X}, \tag{2}$$

where x is the received value, and X is the value given in the project.

**Table 7.** Comparative analysis of the results of the measured and model values of the voltage and current of the GTPS "Ilyichevskaya".

| Object | Parameter | Project Value | Measured Value | Calculated Value | Deviation of the Calculated Value from the Measured Value δ, % |
|---|---|---|---|---|---|
| GTPS "Ilyichevskaya" busbar-1 | Voltage, kV | 6 | 6.28 | 6.108 | 2.7 |
| GTPS "Ilyichevskaya" busbar-2 | | 6 | 6.19 | 6.112 | 1.3 |
| LM-2 | Current, A | 385 | 338.3 | 347.8 | 2.8 |
| LM-3 | | 385 | 299.8 | 303.6 | 1.3 |

The problems discussed above are typical for the oil and gas field's gas turbine power stations. Basic recommendations to ensure the quality parameters of electrical energy are:

- To analyze key modes of gas turbine power stations and to design a map of the operating switches of the GTPS' elements;

- To accompany the most problematic modes with instrumental measurements;
- To consider the possibility of installing filter-compensating devices to reduce the overall level of distortion or other equipment for the compensation of higher harmonic components;
- To consider the possibility of installing dynamic voltage distortion compensators to eliminate problems associated with voltage failures and landings, overvoltages of the PSS.

## 9. Conclusions

The formed mathematical description of the electrotechnical complex of the oil field, taking into account the nonlinearity of the two-shaft GTU, together with the instrumental measurements carried out, made it possible to assess in more detail the impact of the influence of GTPS based on an aviation gas turbine on the quality of electrical energy of the PSS.

A detailed analysis of instrumental measurements made it possible to distinguish the influence of elements of the distribution network of the GTPS "Ilyichevskaya" and the main network, so that it is possible to specify solutions to improve the efficiency of the PSS. Thus, to meet the requirements of standards [28,29], it is recommended to consider the possibility of installing filter-compensating devices and voltage distortion dynamic compensation devices.

In general, it should be noted that parallel operation with the main power system causes a slight decrease in the PQI compared to the GTPS "Ilyichevskaya" local mode, but greater stability to disturbing influences, including load dumping.

The recorded deviations of the PQI and the parameters of the GTPS operating mode will allow us to clarify their possible causes and develop a list of measures to eliminate them in the next stage of work.

The measurement data allowed us to record deviations of the PQI from the normally permissible values in the PSS not only due to power flows from the traction SS, but also due to the presence of non-sinusoidal distortions on the side of consumers.

The results of the impact assessment of the GTPS operation mode on power quality will allow us to assess abnormal equipment overloads caused by the deviation of PQI from the normally permissible values and to develop timely measures to eliminate them, for example, equipment degradation management [37–39]. Moreover, this expands the set of acceptable solutions for control algorithms, taking into account territorial and geological conditions and factors of associated petroleum gas utilization on oil and gas fields.

**Author Contributions:** Conceptualization, A.P. and A.L.; methodology, A.R., D.L. and E.P.; software, D.L., Y.G., R.S. and P.B.; validation, A.K. (Andrei Kokorev) and A.K. (Aleksandr Kokorev); formal analysis, A.P., A.R., D.L., A.L., E.P. and P.I.; writing—original draft preparation, A.P.; writing—review and editing, A.P. and A.R.; visualization, A.R., A.L. and D.L.; supervision, A.P. and A.R. All authors have read and agreed to the published version of the manuscript.

**Funding:** The research was carried out in the organization of the Lead Contractor as part of the R&D carried out with the financial support of the Ministry of Science and Higher Education of the Russian Federation (agreement number 075-11-2021-052 of 24 June 2021) in accordance with the decree of the Government of the Russian Federation: 9 April 2010, number 218 (PROJECT 218). The main R&D contractor is Perm National Research Polytechnic University.

**Institutional Review Board Statement:** Not applicable.

**Informed Consent Statement:** Not applicable.

**Conflicts of Interest:** The authors declare no conflict of interest.

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
