# Peer review of "Investigation of the Influence of Gas Turbine Power Stations on the Quality of Electric Energy in the Associated Petroleum Gas Utilization"

_sustainability, doi:10.3390/su14010299_

Round 1

Reviewer 1 Report

Please make clear reference position No. 1. Draft from 2015?, link is invalid. Please clarify document status.

Author Response

Tnanks a lot for your notice! We clarified the newest link (No1 in References)

Reviewer 2 Report

The article presents an analysis of the impact of gas turbine power station on electrical energy quality in petroleum gas field applcations. The authors present the used mathematical models, the electrical system under study, as well as the power quality measurements of the electrical system.

A comparative study between the possible operation modes and measurements of the electrical system was carried out in order to support the authors' analysis and decisions. Thus, the operation modes of a gas turbine power station that affect the power quality are identified and discusses in the conclusions.

In addition to the analyses, the authors recommend the installation of specific equipment in order to mitigate the problems related to energy quality identified in the study.

Just as a suggestion to improve writing, I suggest correcting the following types:
- In line 125 I believe that the correct should be GTU-2 and not GTU-3.
- In line 126 I believe that the correct should be GTU-3 and not GTU-2.
- In line 333 correct the typo "Busnar".

Author Response

Thanks a lot for your comments!

Fixed incorrect designation of GTU-2 and GTU-3 in lines 126, 127, as well as further in the text (lines 271, 283, 305, 325, 344, 360).
Fixed a typo on lines 333 and 386.

Reviewer 3 Report

The authors reported "Investigation of the influence of gas turbine power stations on the quality of electric energy in the associated petroleum gas utilization ". The manuscript presents a mathematical description of power supply systems and provides solutions on operation modes on power quality between energy consumption of the devices and the energy supply. However, the originality of manuscript is not high.

In the experimental results the authors stated that "We can conclude that the source of a short-term overstepping the normally permissible limits of the harmonic components is modes switching, and the source of longterm fixed disturbances is located in the main power system (cell 11 in Scheme 1)." In order to highlight the message beyond the conclusion, this statement should be detailed and rephrased.

Author Response

The article presents the factors of operation of a gas turbine power station that affect the power quality are identified and recommendations for their elimination are given.

In local mode, the harmonic components in the measurement nodes is practically absent (the values of the harmonic component coefficient are less than 0.2 %). It can be seen that after switching from the local mode of the GTPS "Ilyichevskaya" to the parallel mode with the main network, the coefficient of harmonic components increases significantly (up to 1.5%).

We have made these clarifications in the article. Also we've added in the conclusion, about which you mentioned, the statement:

A detailed analysis of instrumental measurements made it possible to distinguish the influence of elements of the distribution network of the GTPS "Ilyichevskaya" and the main network, so that it is possible to specify solutions to improve the efficiency of the PSS.